# Clinical Significance of B-Type Natriuretic Peptide Levels at 3 Months after Atrial Fibrillation Ablation

**DOI:** 10.3390/diseases9030049

**Published:** 2021-07-01

**Authors:** Sen Matsumoto, Yasuharu Matsunaga-Lee, Masashi Ishimi, Mamoru Ohnishi, Nobutaka Masunaga, Koichi Tachibana, Yuzuru Takano

**Affiliations:** 1JCHO Hoshigaoka Medical Center, Department of Cardiovascular Medicine, Hirakata 573-8511, Japan; masashi_10031986@yahoo.co.jp (M.I.); om1208@zeus.eonet.ne.jp (M.O.); nmas1103@yahoo.co.jp (N.M.); tachibana4123@gmail.com (K.T.); hawk.yzr@gmail.com (Y.T.); 2Division of Cardiology, Osaka Rosai Hospital, Osaka 591-8025, Japan; sumomo0304@gmail.com

**Keywords:** catheter ablation, atrial fibrillation, B-type natriuretic peptide, arrhythmia

## Abstract

The role of B-type natriuretic peptide (BNP) levels as a predictor of arrhythmia recurrence (AR) after atrial fibrillation (AF) ablation remains unclear. In this study, we investigated the association of BNP levels before and 3 months after ablation with the risk of AR. A total of 234 patients undergoing their first session of AF ablation were included (68% male, mean age of 69 years). The cut-off value for discriminating AR was determined based on the maximum value of the area under the receiver operating characteristic (ROC) curve. The impact of BNP levels on AR was evaluated using Cox regression analysis. ROC curve analysis showed that the area under the curve for BNP at 3 months after the procedure was larger (0.714) compared to BNP levels before ablation (0.593). Elevated levels of BNP 3 months after the procedure (>40.5 pg/mL, n = 96) was associated with a higher risk of AR compared to those without elevated levels (34.4% vs. 10.9%, *p* < 0.01). Multivariate Cox regression analysis revealed that elevated BNP levels were associated with an increased risk of AR (hazard ratio 2.43; *p* = 0.014). Elevated BNP levels 3 months after AF ablation were a significant prognostic factor in AR, while baseline BNP levels were not.

## 1. Introduction

The introduction of radiofrequency catheter ablation has broadened the treatment options available for patients with atrial fibrillation (AF), whose management presents a complicated challenge to physicians [1]. In particular, catheter ablation has allowed sinus rhythm maintenance in patients with persistent AF; however, chronicity has been forced.

B-type natriuretic peptide (BNP) is a well-known marker of heart disease; particularly, elevated BNP levels have been reported in patients with AF [2]. Moreover, NT-proBNP has been reported to be a predictor of AF onset after adjustment for known risk factors [3].

Elevated levels of BNP before AF ablation have been shown to strongly predict arrhythmia recurrence after the operation [4,5]. These findings suggest that elevated BNP levels reflect advanced atrial diseases in primary AF patients; however, other factors should be considered, including age, AF duration, and atrial size. In contrast, one study showed that a reduction in BNP levels before ablation is a prognostic marker of post-ablation complications [6]. In this study, BNP levels before and 3 months after the blanking period of AR were compared.

## 2. Materials and Methods

### 2.1. Patients

A total of 234 consecutive patients who visited our hospital between 2013 and 2017 were included. These patients underwent their first session of AF ablation.

### 2.2. BNP Level Measurement

BNP values were recorded upon admission and at 3 months after ablation. Based on a consensus statement [7], a 3-month blanking period was used.

### 2.3. Catheter Ablation

Ablation was performed under deep sedation with propofol and dexmedetomidine. A 20-polar electrode catheter (BeeAT, Japan Lifeline, Tokyo, Japan) was inserted and guided through the right subclavian vein to reach the coronary sinus. Two or three long sheaths were introduced from the right femoral vein to reach the left atrium. AF ablation was performed using an open-irrigated contact-force catheter (TheromoCool Smart-Touch SF, Biosense Webster, Diamond Bar, CA, USA) with an electro-anatomical mapping system (CARTO 3, Biosense Webster). Radiofrequency energy was delivered at 30–35 W and 20–25 W onto the posterior wall near the esophagus. After circumferential bilateral PV isolation, continuous isoproterenol infusion and a rapid adenosine triphosphate injection were administered to induce non-PV triggers. Subsequently, the non-PV triggers were ablated using self-reference mapping techniques reported in a previous study [8]. Additional ablation of the cavotricuspid isthmus, superior vena cava, or left atrial posterior wall was performed at the operator’s discretion.

### 2.4. Follow-Up

After ablation, the prescription of class I and III anti-arrhythmic drugs was deferred due to their potential interference with the evaluation of late arrhythmia recurrence. Patients were scheduled to visit the outpatient clinic at 1, 3, 6, and 12 months after the ablation, and annually thereafter. A 24 h Holter electrocardiogram was performed at 3, 6, and 12 months, and annually thereafter. AR was defined as the occurrence of AF, atrial flutter, and atrial tachycardia lasting for more than 30 s after the 3-month blanking period. If these conditions occurred within the first three months, it was defined as an early recurrence.

### 2.5. Statistical Analysis

Statistical analyses were performed using SPSS software (version 19.0; Chicago, IL, USA). Categorical variables were expressed as percentages and compared using the chi-square test. Continuous variables were expressed as mean ± standard deviation or median (interquartile range) and compared using the Student’s *t*-test or Mann–Whitney U test. The discriminatory accuracy of BNP levels for AR was assessed using the area under the curve (AUC) of the receiver operating characteristic (ROC) curve with logistic regression analysis. The cut-off value of BNP levels that balanced sensitivity and specificity for AR was identified, using ROC analysis, as the point on the curve closest to the upper left-hand corner of the AUC. The patients were then divided into two groups according to the previous BNP cut-off values. AR rates were described using the Kaplan–Meier method and compared using the log–rank test. Cox regression models were used to examine the association between elevated BNP levels and risk of AR. Additionally, the impact of AR was evaluated by subgroup analysis. Statistical significance was defined as *p* < 0.05.

## 3. Results

### 3.1. Baseline Characteristics

The baseline characteristics are summarized in Table 1. Among the 234 patients, four were lost before the follow-up. Among the remaining patients, 47 developed AR during the mean follow-up period of 514 days. Regarding the demographic characteristics, patients with structural heart disease (SHD, including coronary artery disease, dilated or hypertrophic cardiomyopathy, and valvular heart diseases), non-paroxysmal AF, and higher baseline BNP levels had an increased risk of AR. Since BNP levels were not normally distributed, the analysis was performed after logarithmic transformation (ln). In the univariate Cox regression analysis, SHD, non-paroxysmal AF, early recurrence, lnBNP before ablation, and lnBNP at 3 months were significantly associated with an increased risk of AR (Table 2).

### 3.2. BNP Levels before Ablation and BNP Levels at 3 Months

The ROC analysis showed that the AUC for BNP levels 3 months after the procedure was significantly larger compared to BNP levels before the procedure (0.714 and 0.593, respectively; *p* < 0.05, Figure 1). At 3 months post-ablation, the optimal cut-off value of BNP levels was approximately 40.5 pg/mL (sensitivity, 0.681; specificity, 0.665). It was found that patients who were older, had increased incidence of SHD and non-paroxysmal AF, higher CHADS2 scorer, larger left atrial dimensions, lower estimated glomerular filtration rate, and elevated BNP levels before the procedure demonstrated elevated BNP levels 3 months after the procedure (Table 3).

### 3.3. BNP Levels at 3 Months and Outcome

A higher risk of AR was observed in patients with elevated BNP levels 3 months after the procedure compared to those without elevated levels (34.4% vs. 10.9%, *p* < 0.01). Furthermore, Kaplan–Meier curves showed that patients with elevated BNP levels 3 months after the procedure had a significantly higher risk of AR (Figure 2). Multivariate Cox regression analysis revealed that elevated BNP levels at 3 months and early recurrence were significant predictors of AR (Table 4). The outcomes of the subgroup analysis are shown in Table 5.

## 4. Discussion

We aimed to determine the association of elevated BNP levels before and after AF ablation with the development of AR. We found that elevated levels of BNP before and 3 months after ablation were associated with an increased risk of AR. Furthermore, the AUC of BNP levels after the blanking period was larger compared to before ablation.

### 4.1. BNP as a Marker

No consensus has been reached regarding the use of baseline BNP levels as a predictor for AR after AF ablation. There are studies reporting that elevations [4,5] or reductions [6] in BNP levels are predictors of AF; however, Pellarisetti et al. [9] stated that BNP levels were neither a short- or long-term predictor of AR. In this study, elevated BNP levels 3 months after the procedure was more useful as a prognostic marker and predictor of recurrence compared to baseline BNP levels. The cardiac rhythm was not comprehensively examined during blood collection; however, most patients with PAF had a normal sinus rhythm on admission. Blood sample collection and Holter electrocardiography were performed in succession; to obtain similar results, only cases with a normal sinus rhythm on Holter electrocardiograms were analyzed. The optimal cut-off value of BNP levels at 3 months showed low sensitivity (0.681) and specificity (0.665). It is unknown whether the results of the study population are representative of real-world data. We believe measuring BNP levels is warranted in patients who are suspected of recurrence. Therefore, BNP level is not an absolute marker of AR; however, its use as a predictor of recurrence should be considered.

### 4.2. AF and Elevated BNP Levels

Elevated BNP levels might contribute to changes in left ventricular filling pressure because AF is associated with the dyssynchronization of atrium mechanics and activation of the sympathetic nervous system [3,10]. A report has shown that in patients with AF, BNP is produced in the atrium [11]. However, the electrophysiological effects of BNP remain unclear. Springer et al. [12] showed that under basal conditions, BNP exerted no effect on atrial myocytes; however, BNP increased atrial action potential duration and L-type Ca^2+^ current in the presence of isoproterenol. Therefore, it is possible that AF recurrence caused elevations in reduced BNP levels.

### 4.3. Involvement of Heart Disease in Elevated BNP Levels

A recent report of patients with heart failure (HF) showed a lower risk of mortality or hospitalization after AF ablation [13]. In the current study, most patients had more than 40% of LVEF measured by the modified Simpson’s rule and Teichholz method using transthoracic echocardiography, and only 2.6% of the enrolled patients exhibited LV dysfunction. Considering this, the prescription rate for diuretics was low; therefore, it did not affect BNP levels (data not shown). In non-paroxysmal AF patients, the elevated BNP value might predict AR as compared to those with paroxysmal AF, and pathological conditions such as atrial cardiomyopathy could be considered (Table 5).

### 4.4. Clinical Implication

A recent meta-analysis demonstrated that long-term alleviation from atrial arrhythmia requires multiple procedures [14]. We showed the effectiveness of BNP levels as a prognostic factor for AR, considering follow-up data compared to the baseline. In addition, for cases with elevated BNP levels, it may be necessary to perform a 24 h or 7-day Holter electrocardiogram to detect asymptomatic AF. Finally, recent reports have shown that AF ablation is associated with a lower incidence of stroke and death [15]. In our opinion, patients with suspected recurrence should be given focus; furthermore, it can be used as a determinant of discontinuing anticoagulant therapy.

### 4.5. Study Limitations

This was an observational retrospective study; the data were gathered from a single center from a small sample size. However, a strength of this study lies in its inclusion of patients whos’ treatment was consistent with daily clinical practice. Furthermore, the ablation techniques and technology were performed by different individuals which may affect outcomes [16]. Additionally, inflammatory markers were not measured, including interleukin-6 and C-reactive protein [17]. Further evaluations of the findings may be necessary using cohorts.

## 5. Conclusions

Elevated BNP levels 3 months after the blanking period might be a more significant prognostic factor in AR compared to baseline BNP levels.

## Figures and Tables

**Figure 1 diseases-09-00049-f001:**
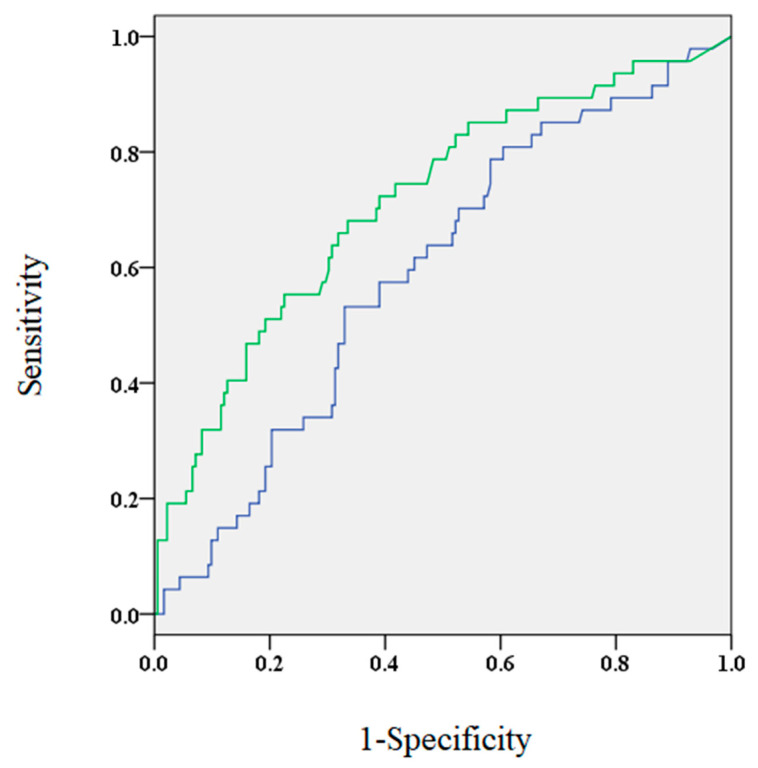
The area under the curve for brain natriuretic peptide (BNP) levels 3 months after the procedure (green) and before the procedure (blue) were 0.714 and 0.593, respectively.

**Figure 2 diseases-09-00049-f002:**
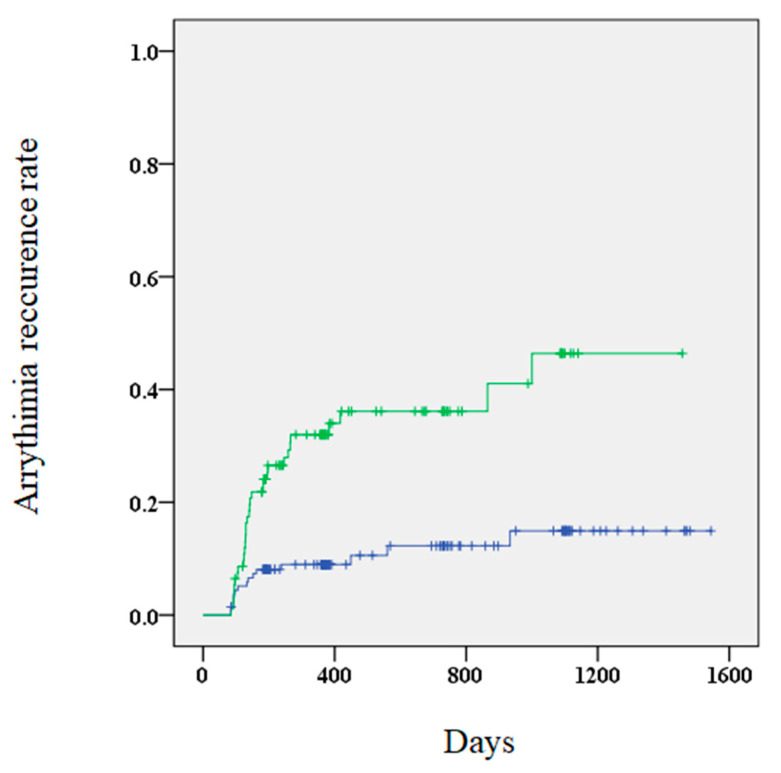
A Kaplan–Meier curve showing the arrhythmia recurrence rate for patients with (green) and without (blue) elevated BNP (brain natriuretic peptide) levels.

**Table 1 diseases-09-00049-t001:** Patient characteristics.

Variables	All	AR (+)	AR (−)	*p* Value
	n = 234	n = 47	n = 183	
Age, years	69 ± 9	70 ± 9	69 ± 10	0.660
Men, n (%)	160 (68.4)	31 (66.0)	127 (69.4)	0.725
Structual heart disease, n (%)	69 (30.0)	20 (42.6)	47 (26.3)	0.033
Hypertension, n (%)	146 (62.4)	25 (53.2)	117 (63.9)	0.183
Diabetes mellitus, n (%)	47 (20.1)	11 (23.4)	35 (19.1)	0.541
CHADS_2_ score	1.74 ± 1.29	1.68 ± 1.20	1.75 ± 1.31	0.754
Atrial fibrillaiton type				
Paroxysmal, n (%)	119 (50.9)	17 (36.2)	100 (54.6)	0.003
Persistent, n (%)	77 (32.9)	15 (31.9)	61 (33.3)	
Long-standing persistent, n (%)	38 (16.2)	15 (31.9)	22 (12.0)	
Echocardiography				
Left ventricular ejection fraction, %	62.9 ± 9.1	61.3 ± 8.3	63.4 ± 9.2	0.164
Left atrial dimension, mm	39.7 ± 6.5	39.9 ± 5.8	39.4 ± 6.6	0.598
Laboratory data				
eGFR, mL/min/1.73 m^2^	61.8 (52.4, 74.6)	61.1 (51.9, 71.0)	62.1 (52.4, 75.1)	0.523
BNP before ablation, pg/mL	78.6 (25.9, 143.1)	119.4 (55.9, 159.7)	72.6 (23.5, 138.6)	0.049

BNP, brain natriuretic peptide; eGFR, estimated glomerular filtration rate; AR, arrhythmia recurrence; data presented as %, mean ± SD or median (25th–75th percentile).

**Table 2 diseases-09-00049-t002:** Factors associated with arrhythmia recurrence after AF ablation.

	Unadjusted		
Variable	HR	95%CI	*p* Value
Age, +1 y	1.01	0.98–1.04	0.577
Men	0.88	0.48–1.61	0.672
SHD	2.00	1.12–3.56	0.020
CHADS_2_ score > 1	1.09	0.61–1.93	0.771
Non-PAF vs. PAF	2.01	1.11–3.65	0.022
LVEF, +1%	0.98	0.95–1.01	0.131
LAD, +1 mm	1.02	0.98–1.07	0.356
Early recurrence	5.22	2.93–9.30	<0.001
ln BNP before ablation	1.36	1.04–1.77	0.026
ln BNP at 3 months	2.09	1.59–2.74	<0.001

AF, atrial fibrillation; CI, confidence interval; HR, hazard ratio; LVEF, left ventricular ejection fraction; LAD, left atrial dimension; PAF, paroxysmal AF; SHD, structural heart disease; PAF, paroxysmal AF.

**Table 3 diseases-09-00049-t003:** Comparison of patient characteristics between patients with elevated and low BNP levels.

Variables	All	Elevated BNP	Low BNP	*p* Value
	n = 234	n = 96	n = 138	
Age, years	69 ± 9	72 ± 8	67 ± 10	<0.001
Men, n (%)	160 (68.4)	59 (61.5)	101 (73.2)	0.064
Structural heart disease, n (%)	69 (30.0)	39 (41.1)	30 (22.2)	0.003
Hypertension, n (%)	146 (62.4)	61 (63.5)	85 (61.6)	0.785
Diabetes mellitus, n (%)	47 (20.1)	21 (21.9)	26 (18.8)	0.620
CHADS2 score	1.74 ± 1.29	2.07 ± 1.34	1.50 ± 1.20	0.001
Atrial fibrillation type				
Paroxysmal, n (%)	119 (50.9)	38 (39.6)	81 (58.7)	0.002
Persistent, n (%)	77 (32.9)	34 (35.4)	43 (31.2)	
Long-standing persistent, n (%)	38 (16.2)	24 (25.0)	14 (10.1)	
Echocardiography				
Left ventricular ejection fraction, %	62.9 ± 9.1	63.6 ± 9.0	62.4 ± 9.2	0.306
Left atrial dimension, mm	39.7 ± 6.5	41.1 ± 6.4	38.7 ± 6.5	0.006
Laboratory data				
eGFR, mL/min/1.73 m^2^	61.8 (52.4, 74.6)	56.3 (46.9, 67.9)	63.9 (55.8, 77.2)	<0.001
BNP before ablation, pg/mL	78.6 (25.9, 143.1)	127.8 (72.2, 221.0)	47.8 (16.3, 110.8)	<0.001

BNP, brain natriuretic peptide; eGFR, estimated glomerular filtration rate; data presented as %, mean ± SD or median (25th–75th percentile).

**Table 4 diseases-09-00049-t004:** Factors associated with arrhythmia recurrence after AF ablation with adjusted values.

	Adjusted		
Variable	HR	95%CI	*p* Value
Age, +1 y	0.99	0.96–1.03	0.734
Men	1.35	0.68–2.69	0.385
SHD	1.60	0.83–3.12	0.163
CHADS_2_ score > 1	0.80	0.41–1.56	0.513
Non-PAF vs. PAF	1.24	0.61–2.50	0.554
LVEF, +1%	0.98	0.95–1.01	0.208
LAD, +1 mm	1.00	0.95–1.06	0.970
Early recurrence	4.51	2.36–8.63	<0.001
elevated BNP levels	2.43	1.20–4.91	0.014

AF, atrial fibrillation; CI, confidence interval; HR, hazard ratio; LVEF, left ventricular ejection fraction; LAD, left atrial dimension; PAF, paroxysmal AF; SHD, structural heart disease; PAF, paroxysmal AF.

**Table 5 diseases-09-00049-t005:** Subgroup analysis of arrhythmia recurrence.

	Hazard Ratio	95%CI	*p* for Interaction
Age			0.788
≥75 yr	4.99	1.38–18.0	
<75 yr	3.56	1.73–7.34	
Sex			0.583
Men	3.18	1.54–6.56	
Women	5.27	1.50–18.6	
SHD			0.637
Yes	2.70	0.98–7.42	
No	3.82	1.74–8.37	
CHADS_2_ score			0.213
≥2	6.62	2.22–19.7	
<2	2.66	1.17–6.03	
AF type			0.009
Paroxysmal	1.20	0.44–3.24	
Non-paroxysmal	8.97	3.11–25.9	
LVEF			0.858
<60%	4.04	1.60–10.2	
≥60%	3.74	1.63–8.55	
LAD			0.333
≥40 mm	5.26	1.93–14.3	
<40 mm	2.77	1.23–6.25	
Early Recurrence			0.550
Yes	3.21	1.10–9.38	
No	2.07	0.88–4.88	
eGFR			0.768
<60 mL/min/1.73 m^2^	4.27	1.57–11.6	
≥60 mL/min/1.73 m^2^	3.45	1.54–7.72	

AF, atrial fibrillation; CI, confidence interval; eGFR, estimated glomerular filtration rate; LVEF, left ventricular ejection fraction; LAD, left atrial dimension; SHD, structural heart disease.

## Data Availability

The data presented in this study are available in this article.

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
