# Peer review of "Clinical Significance of B-Type Natriuretic Peptide Levels at 3 Months after Atrial Fibrillation Ablation"

_diseases, 2021, doi:10.3390/diseases9030049_

Round 1

Reviewer 1 Report

Dear authors of the titled work: "Clinical significance of B-type natriuretic peptide levels at 3 2 months after atrial fibrillation ablation", It seems to me that his work is original, it has a very good statistical analysis, the material and method used very well and they reach good conclusions, aligning the objectives with the results and conclusions. Only that I would add the most up-to-date bibliography of the last two years, such as the FA 2020 European guides or some updated bibliographies extracted from this guide

Author Response

We thank you very much for your careful review of our manuscript.

Reviewer 2 Report

  1. By which method was BNP determined BNP? Which value was considered cut off?
  2. By which method was the left ventricular ejection fraction calculated?
  3. Were patients with heart failure excluded from the study? The presence of this condition could influence the values of BNP.
  4. Nothing is specified (only in passing under the heading "Discussions") about the treatment of associated cardiac disease, given that some drugs may influence the level of BNP.

Author Response

Reviewer #2:
We thank you very much for your careful review of our manuscript. We have tried to incorporate all the suggestions made, and believe that these suggestions have significantly improved the content of this manuscript. The following are the detailed descriptions of the changes that have been made:

1) By which method was BNP determined BNP? Which value was considered cut off?

Response: Thank you for pointing this out. The cut-off point of BNP levels to detect arrhythmia recurrence was identified using ROC analysis. That is, we think that the best cut-off value provides both the highest sensitivity and the highest specificity, easily located on the ROC curve by finding the highest point on the vertical axis and the furthest to the left on the horizontal axis. We added as follows.

The cut-off point of BNP levels that balanced sensitivity and specificity for AR was identified using ROC analysis as the point on the curve closest to the upper left-hand corner of the AUC to detect it. (Page 2, Lines 76-78)

2) By which method was the left ventricular ejection fraction calculated?

Response: Thank you for pointing this out. We measured the LVEF by modified Simpson method based on biplane method of disks and Teichholz method based on simple ellipsoid shape with a correction factor. We added as follows.

In the current study, most patients had more than 40% of LVEF measured by modified Simpson’s rule and Teichholz method using transthoracic echocardiography, and only 2.6% of the enrolled patients exhibited LV dysfunction. (Page 7, Lines 165-167)

3) Were patients with heart failure excluded from the study? The presence of this condition could influence the values of BNP.

Response: As you said, patients with acute decompensated heart failure were excluded. Most patients were scheduled to be hospitalized.

4) Nothing is specified (only in passing under the heading "Discussions") about the treatment of associated cardiac disease, given that some drugs may influence the level of BNP.

Response: Thank you for pointing this out. We think you're right. The purpose of this study was to examine the evaluation of ablation treatment effect in patients who underwent ablation for atrial fibrillation. In addition, ablation was performed on the indication cases according to the guidelines at that time, and complicated cases were excluded as shown in the patient backgrounds. Therefore, the prescription rate of drugs that affect BNP such as diuretics and beta-blockers was low, and we excluded them from the analysis, and did not consider them.
